# Analysis of entropy on the European markets of energy and energy commodities prices

**Daniel Papla** ⓘ *, **Rafał Siedlecki**

Wroclaw University of Economics and Business, Wrocław, Poland

* daniel.papla@ue.wroc.pl

## Abstract

The paper analyzes the problem of entropy in the moments of transition from a normal economic situation (2015–2019) to the Pandemic period (2020–2021) and the period of Russia's attack on Ukraine (2022–2023). The research in the article is based on the analysis of electricity, oil, coal, and gas prices in 27 countries of the European Union and Norway. The daily data cover the period from January 1, 2015, to March 30, 2023, and were analyzed using two-dimensional sets of electricity and commodity prices. The work uses the time dependent James-Stein estimator of the Shannon informational entropy.

**Data Availability Statement:** All relevant data are within the manuscript and its Supporting Information files.

**Funding:** The author(s) received no specific funding for this work.

## 1. Introduction

Issues with energy price analysis and its forecasts are known but still current and interesting. In today's "turbulent" times, there is no doubt that forecasting electricity prices is not only for participants of commodity and derivative markets (day traders and speculators) but also for energy systems planning and operations [1, 2].

In the literature, they are often analyzed using statistical and econometric methods to describe financial market data, where either a normal distribution of returns and their "stationarity" is assumed or variability of variance is analyzed [3–5]. In practice, attempts are made to make time-series stationary by determining logarithmic rates of return, which only "reduces" their non-stationarity but does not eliminate it [5]. The literature often notes that research based on these methods or artificial neural networks (ANN) leads to difficulties in their application, and forecasts are often acceptable only once or for a short period (they are not universal) [6, 7]. It is because the prices in the energy market are characterized by very high volatility and outliers caused by many factors, both measurable and unmeasurable: weather conditions, economic and political factors, technology development and problems with its storage, among others. So those prices can be treated as a probabilistic values [1, 8]. Prices and consumption are also influenced by the emergence of new energy sources (renewable energy; green energy) and the variability (fashion) of using old ones, such as e.g. nuclear energy. The use and sharing of these sources in overall consumption varies depending on countries and continents (see Fig 1). In European countries, the United States and Canada, there is a significant decrease in the share of coal in energy sources, while in Asia, it rapidly increases, making it the main energy source in the world, along with gas and oil.

Currently, we are dealing with significant instability not only in the financial and commodity markets (e.g. energy) but also in the economy and social policy (ecology), caused

**Competing interests:** The authors have declared that no competing interests exist.

**Fig 1. Primary energy consuption by source.** Source: Energy Institute Statistical Review of World Energy (2023).

by, among others, the COVID-19 pandemic and the armed conflict in Ukraine and the Middle East, but also the very rapid development of new technologies such as A.I. leading to high energy demand. In times without turbulences and shocks, the analysis of prices on financial markets (e.g. stock exchanges) and energy indicated the possibility of forecasting long-term trends and the lack of random walks for daily or monthly observations [3, 7].

In this contribution, the entropy analysis as a statistical method is used to analyze energy prices in markets. The concept of entropy occurs primarily in thermodynamics but also the theory of probability, information, stochastic processes, and economics and finance [9–11].

As in [12] "Thermodynamics as this drives the world economy from its ecological foundations as solar energy passes through food chains in dissipative process of entropy rising and production fundamentally involving the replacement of lower entropy energy states with higher entropy ones". Thus, tools and methods for measuring entropy can effectively model and analyze financial and commodity markets. Based on financial market theory, prices of energy drivers such as coal, oil, or gas are similar to the prices of financial assets during "normal" times following a random walk. Therefore, it can be concluded that the phenomenon of energy price drivers is an isolated system that tends to maximum entropy more or less slowly. In finance and economics, entropy often determines the measure of system order (which can be defined as moments of the high degree of market efficiency), which is a prerequisite for making effective forecasts [3, 12]. It means there is a high entropy of financial asset prices, which can lead to severe mistakes made by financial analysts and economists, which can deepen the crisis. According to Bejan Constructal Law [13]: "For a finite-size flow systems to persist in time, it must evolve with freedom such it provides an easier access to its flows". The paper aims to investigate whether in crisis situations, in which external interference occurs, entropy decreases, which means that the possibility of forecasting the prices of electricity and its carriers (namely oil, coal and gas) increases.The following hypothesis was put forward in the article: *Similar to the second law of thermodynamics*, *during crises*, *external interference in the market (e.g., price regulation) causes decrease of the entropy of energy prices and the factors affecting energy prices are becoming more predictable.* According to the principles of physics, in an isolated system, entropy increases or decreases at the moment of external intervention [14, 15], similar to finance, where during crises and economic slowdowns, there is interference from governments introducing new regulations and intervening in financial markets [6, 16]. Our research support the hypothesis that regulations make market behavior less random.

To verify hypothesis, the Shannon entropy with an estimator by James-Stein [17] is used. The method allows us to take into account the non-linear nature of the studied phenomena and also analyze current data, from the period of the COVID-19 pandemic and from the current period, including, among others, armed conflict in Ukraine. This method seems to be best adapted to our case, since it does not require the analysis of the stationarity of the series, it is simple and requires a small number of assumptions, which makes this method attractive and can be applied even without deep knowledge of statistics and econometrics. Since this method does not require the analysis of the stationarity of the series, it is simple and requires a small number of assumptions, which makes this method attractive and can be applied even without deep knowledge of statistics and econometrics. This entropy estimator also performs very well in the small sample situation [17]. Statistical methods assume normality of series and stationarity and therefore often the results are on the borderline. When studying prices it would be necessary to assume and exclude the trend and assume a normal distribution. Return rates also do not have a normal distribution. The method seems to be the best suited not necessarily the most effective. The simple method may be less effective but in the long run requires less interference in the model.

For entropy estimation, especially for time-dependent entropy, we used proprietary software created in the R package for the paper. A robustness analysis of the results is also conducted, we have calculated the entropy estimator for the daily data with one year moving window and for the monthly data with two years moving window and both results corroborates our original results. Those results are included in the appendix.

## 2. Methodology

Entropy is a measure of the degree of uncertainty associated with the variables used in statistical phenomena in the context of statistical systems or probability theory.

In order to numerically quantify the amount of "lost information" in phone-line signals, C. E. Shannon proposed a measure of uncertainty in 1948 that was later dubbed Shannon entropy. Based on works by Nyquist [18, 19] and Hartley [20], the measure was first presented in his well-known work A Mathematical Theory of Communication [21]. It was a key component in the development of information theory, the first comprehensive mathematical theory of communication. Shannon made a major contribution when he demonstrated that entropy could be applied to any series in which probabilities exist. This was a major advancement over earlier research by Clausius and Boltzmann, which was limited to thermodynamic series. The average quantity of "information, choice, and uncertainty" encoded in patterns extracted from a signal or message is what Shannon formally defined as entropy. According to some interpretations, entropy is a system's degree of disorder and unpredictability. The application of Shannon's entropy to any series with a distinct probability distribution was recognized early on and was extensively used, especially in the financial science.

We can define the Shannon entropy [21], using a categorical random variable with corresponding cell probabilities (frequencies) $p_1, \ldots, p_n$ and alphabet size $n$, where $p_k > 0$ and $\Sigma_k p_k = 1$. In our assumption $n$ is fixed and known. The Shannon entropy in natural units of information [21], i.e. units of information based on natural logarithms, is provided by:

$$H = -\sum_{k=1}^{n} p_k ln(p_k)$$

Since the underlying frequencies known as values of the probability mass function are unknown in practice, it is necessary to estimate $H$ and $p_k$ from observed cell counts $y_k \geq 0$.

The maximum likelihood (ML) estimator, which is created by plugging the ML frequency estimates into Shannon equation, is an especially straightforward and popular entropy estimator.

$$\hat{p}_k^{ML} = \frac{y_k}{m}$$

with $m = \sum_{k=1}^{n} y_k$ being the total number of counts.

After some consideration, we have chosen the James-Stein shrinkage estimator [17], which is well adapted to our case because it performs better for data with less information, that is where the number of observations (in our case 250 for one year and 500 for two years window) is small comparing with the number of cells (in our case 100 x 100 = 10 000). This estimator is based on the weighted average of two models: a high-dimensional model with low bias and high variance–the maximum likelihood estimator—and a lower-dimensional model with larger bias but smaller variance–the shrinkage target [17]:

$$\hat{p}_k^{Shrink} = \lambda t_k + (1 - \lambda)\hat{p}_k^{ML}$$

where $\lambda \in [0, 1]$ is the shrinkage intensity, and $t_k$ is the shrinkage target that ranges from 0 (no shrinkage) to 1 (complete shrinkage). The uniform distribution of $t_k = \frac{1}{n}$ is a practical option [17].

Of course there are also other forms of entropy like the Tsallis entropy [22–24] or the Rényi entropy [25–28], but for our research the Shannon entropy seems the best choice. Unlike the Tsallis and Rényi entropy, it was created from the outset with application to information theory, and the other two were created more with reference to physics (chemistry,

thermodynamics and even quantum physics). Of course, it will be very interesting to adapt and modify other entropy methods to finance.

A common presumption in many statistical techniques is stationarity. It ensures that under time translations, processes' mean, variance, and auto-correlation structure is invariant. For the purpose of modeling and forecasting financial systems, this is particularly crucial. However, in the world of finance, series are usually nonstationary, which means that their statistical characteristics vary over time and they are unpredictable models with drifts and trends. Mathematical transformations like logarithms and first differences can make time series almost stationary, making them suitable for use with conventional statistical techniques such as ARMA or GARCH models. Time-dependent methods become good substitutes when transformations to stationary processes are not feasible or possible, for example even after the first differentiation of some processes remain to be nonstationary. To produce a temporal entropy evolution, for example, time-dependent entropy (TDE) methods have been introduced in information theory. TDE is able to capture changes in local irregularity in a signal, which can provide valuable information about its nonstationary dynamics [29]. Applications come from a variety of fields, including finance and physics [30–35].

The time-dependent entropy method is defined formally as follows. Considering a nonstationary time series $Z = z_1, \ldots, z_n$ with time-varying statistical characteristics, time-varying information cannot be captured by standard entropy methods. We define sliding window $Z_t = z_{1+t\Delta}, \ldots, z_{w+t\Delta}$ of size $w \leq N$ at each time step $t = 0, 1, \ldots, \left[\frac{N}{w}\right] + (w - \Delta - 1)$ with a sliding step of $\Delta \leq w$. The operator [.] denotes casting the argument into an integer. A temporal evolution of entropy is produced by computing the desired entropy at a specific time t using the values of the time series in each window $Z_t$. It is well known that variables like window size and time lag can impact outcomes. A few studies using EEG signals are displayed in [36].

## 3. Results

The application proposed in this paper is based on the analysis of electricity, oil, coal, and gas prices in 22 countries of the European Union and Norway (associated countries). Data was obtained from Our World in Data, Eurostat, and Reuters databases. Daily data from January 1, 2015, to March 30, 2023 are used. To check the robustness of our results we also used monthly data.

For each country, three two-dimensional sets of the country's electricity price versus oil, coal, and gas prices are constructed. For each set and each country, time-dependent Shannon entropy [21] is estimated using, as stated earlier, the James-Stein shrinkage estimator [17], with a two-year window, and the step is one day. To get cumulated results for Europe, the average of the estimated entropy for each day is calculated.

According to the research hypothesis, main energy commodities prices in relation to electricity prices in the period 2015–2019 were unpredictable (see Figs 2 and 3). A high level of entropy (random walk phenomenon) was visible in every analyzed country. The decline during armed conflict in Ukraine was less pronounced in such countries like Denmark or Sweden. Throughout Europe, the price of coal had the greatest impact on the price of electricity despite an evident decline in its consumption in Europe.

During the pandemic, the decrease in entropy was small, so the prognostic abilities did not change despite the crisis and turbulence in the energy market in some countries. The time of entropy decline was March 2022, i.e. the beginning of the conflict in Ukraine and the slow abandonment of gas, oil and coal of Russian origin. It was political and legal interference (into the closed system of production and distribution of the electricity market), mainly in European countries. The greatest decline was in the case of post-communist countries, i.e. those most at risk and, at the same time, dependent on gas and oil from Russia.

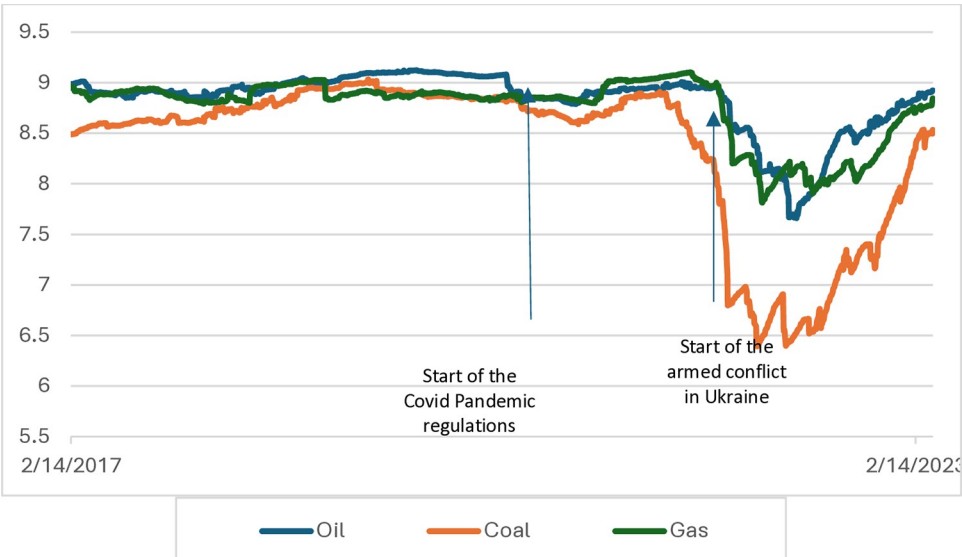

**Fig 2. Time-dependent entropy in Europe–two-year window (maximum entropy ratio 9.21).** Source: own calculations.

Our research confirms previous research [37–39] stating, that during the crisis, when governments support financial markets, entropy decreases, and during stable times, entropy is close to the maximum, and, due to random behaviour of the prices, there could be significant problems with short-term forecasts [40].

In the near future, on the one hand, a transition to green energy and a decrease in demand for oil, coal, and gas as a source of electricity is expected; on the other hand, dynamic technological development significantly increases the electricity demand, which in turn will extend the transition time to green energy. It can cause another period of increased entropy and a decrease in predictability.

Results from monthly data with two year window and daily data but with one year window (see Figs 4 and 5) mostly corroborate our results, we can see the same sharp decline in entropy in March 2022, so we can say that our method is robust in case of frequency of data and window length. Only due to the lower frequency of the data, differences are visible in the form of a temporary increase in entropy and subsequent decline at the beginning of 2023.

## 4. Conclusions

In finance, statistical and econometric methods are most often used to analyze relationships and "behaviours", which require knowledge of the probability distribution and, in the case of prices on financial markets, most often the normal distribution. In our approach, we did not seek the distribution of prices as we had detailed information about them. We have calculated the entropy of systems in different periods.

Methods used in physics and engineering sciences can be used in finance and economics because there are many analogies with phenomena existing in physics, such as Constructal Law and the 2nd law of thermodynamics (entropy, Verhulst and Bejan curve) [10] market temperature and price volatility on financial markets or analysis of business cycles and economic forecasts. This is also confirmed by the Nobel Prize winners' Black-Sholes model, which is derived from the analysis of physical processes and used to value derivatives. We also confirm that one may look for some analogies with Constructal law and entropy in finance and

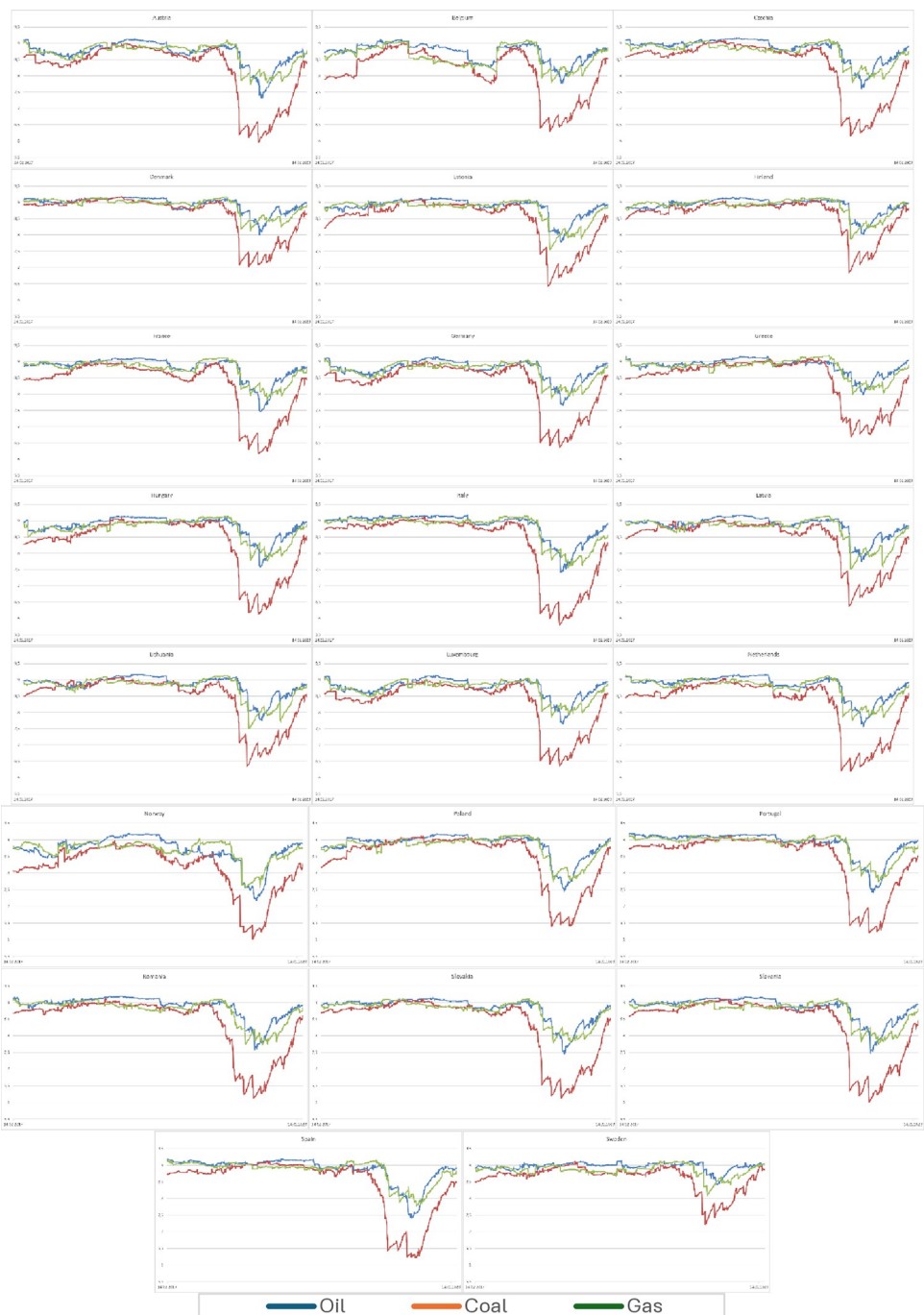

**Fig 3. Time-dependent entropy in chosen European countries–two-year window (maximum entropy ratio 9.21).**
Source: own calculations.

economics. We may suspect that after some time, the free market should reach some "equilibrium state" characterized by a "random walk" of investigated quantities, particularly prices. In such a case, the price would be an analogue of velocity, and thus, some function of prices should correspond to the total energy of the system. Our research confirms the above theories on the example of prices of primary energy sources (oil, gas or coal) in the EU-27 countries,

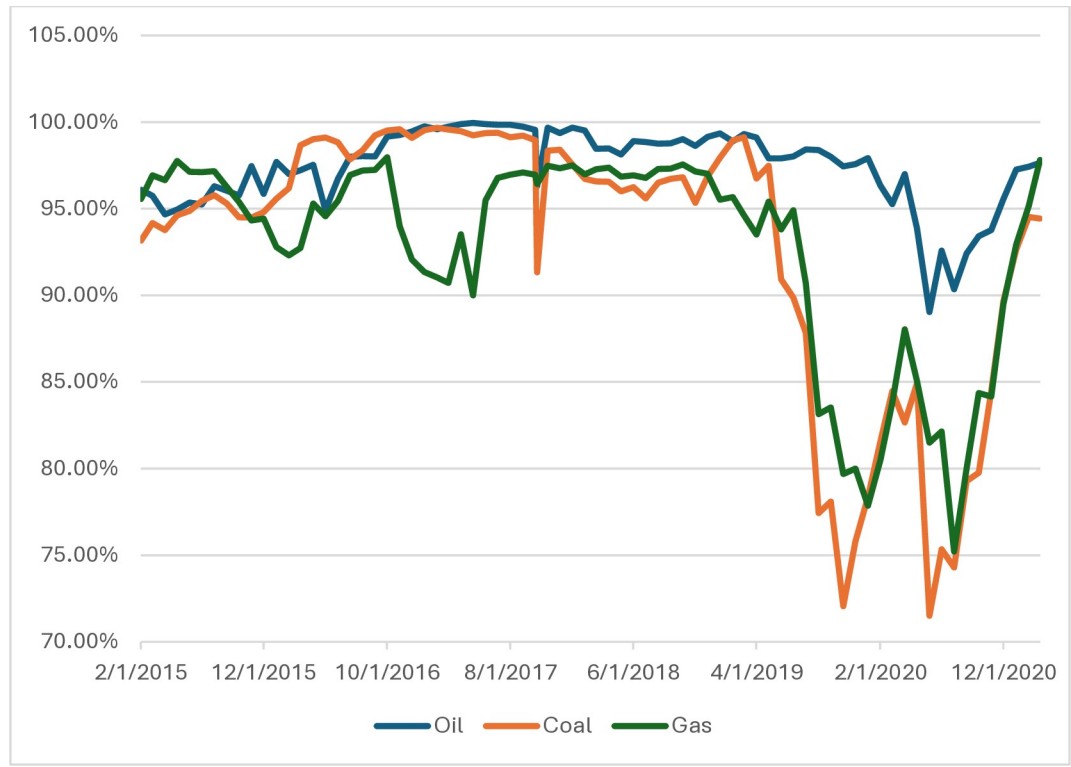

**Fig 4. Time-dependent entropy as a percentage of maximum entropy in Europe–two-year window, monthly data.**
Source: own calculations.

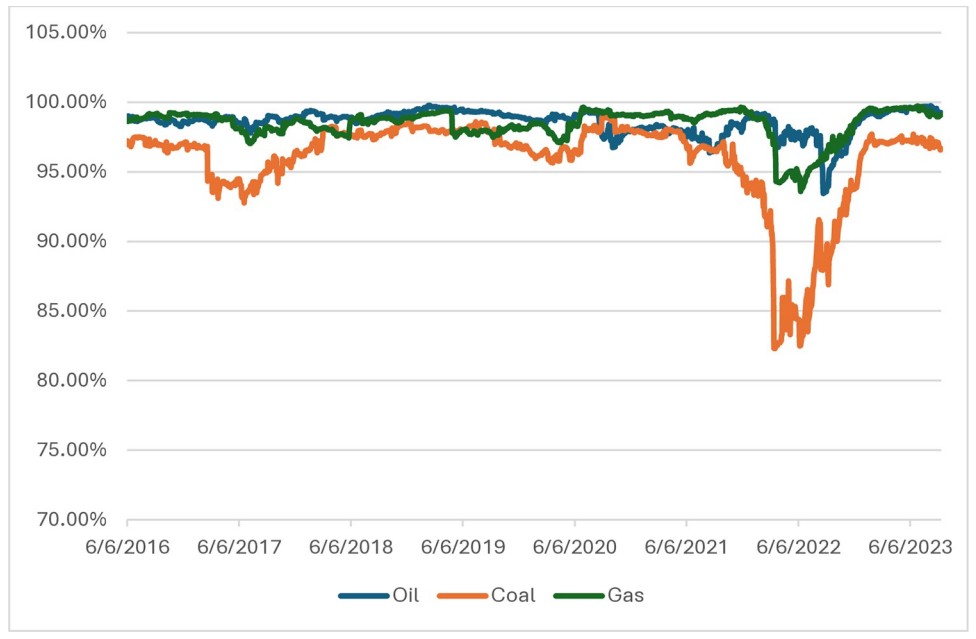

**Fig 5. Time-dependent entropy as a percentage of maximum entropy in Europe–one-year window, daily data.**
Source: own calculations.

i.e. the poor ability to forecast prices in normal conditions and an increase in forecasting possibilities and the impact of other factors on electricity prices in times of crisis, when "outside" actions are taken (e.g. government or international institutions' interventions). Entropy did not drop during the pandemic, i.e. the shock on the financial and commodity markets, but only after Russia attacked Ukraine and the interference of European governments in the energy market, which resulted in more remarkable forecasting ability. However, this was temporary, and, as in isolated systems, entropy began to increase despite prolonged government regulations, returning to the level before the attack (March 2022).

Electricity prices, like prices of financial instruments in short periods, are subject to random walk, i.e. high entropy, which makes building effective econometric and statistical models much more difficult and often impossible.

Of course more statistical approach, especially confidence analysis and comparison of results using other methods, such as the Hurst coefficient, econometric methods (ARIMA, GARCH, etc.) and different methods of examining entropy is in the course of our research. We hope it will be presented in subsequent publications.

## Supporting information

**S1 Data.**
(XLSX)

**S2 Data.**
(XLSX)

**S3 Data.**
(XLSX)

## Author Contributions

**Conceptualization:** Daniel Papla, Rafał Siedlecki.

**Data curation:** Daniel Papla, Rafał Siedlecki.

**Formal analysis:** Daniel Papla, Rafał Siedlecki.

**Funding acquisition:** Rafał Siedlecki.

**Investigation:** Daniel Papla, Rafał Siedlecki.

**Methodology:** Daniel Papla, Rafał Siedlecki.

**Project administration:** Rafał Siedlecki.

**Resources:** Daniel Papla, Rafał Siedlecki.

**Software:** Daniel Papla.

**Supervision:** Rafał Siedlecki.

**Validation:** Daniel Papla, Rafał Siedlecki.

**Visualization:** Daniel Papla, Rafał Siedlecki.

**Writing – original draft:** Daniel Papla, Rafał Siedlecki.

**Writing – review & editing:** Daniel Papla, Rafał Siedlecki.

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
