## [Decision Letter · Decision Letter 0]

30 Sep 2024

PONE-D-24-36364Analysis of Entropy on the European Markets of Energy and Energy Commodities PricesPLOS ONE

Dear Dr. Papla,

Thank you for submitting your manuscript to PLOS ONE. After careful consideration, we feel that it has merit but does not fully meet PLOS ONE’s publication criteria as it currently stands. Therefore, we invite you to submit a revised version of the manuscript that addresses the points raised during the review process.

The authors need to make significant improvements to their article in various areas. The reviewers have provided extensive and detailed advice, and I strongly recommend that the authors follow their suggestions to improve the quality of the article.

We look forward to receiving your revised manuscript.

Kind regards,

Alessandro Mazzoccoli, Ph.D.

Academic Editor

PLOS ONE

Journal Requirements:

2. In the online submission form, you indicated that all relevant data are available from tha Authors on request.

3. Please ensure that you refer to Figure 4 and 5 in your text as, if accepted, production will need this reference to link the reader to the figure.

Additional Editor Comments :

The article, in its current state, is not publishable on PLOS ONE as it needs improvement in several areas. I recommend that the authors follow the reviewers' suggestions.

Reviewers' comments:

Reviewer's Responses to Questions

**Comments to the Author**

1. Is the manuscript technically sound, and do the data support the conclusions?

Reviewer #1: Yes

Reviewer #2: Partly

2. Has the statistical analysis been performed appropriately and rigorously? 

Reviewer #1: I Don't Know

Reviewer #2: Yes

3. Have the authors made all data underlying the findings in their manuscript fully available?

Reviewer #1: No

Reviewer #2: Yes

4. Is the manuscript presented in an intelligible fashion and written in standard English?

Reviewer #1: Yes

Reviewer #2: Yes

5. Review Comments to the Author

Reviewer #1: The paper addresses issues of volatility and extreme outliers of energy prices as impediments to forecasting electricity prices and planning and operating electricity infrastructures. A particular focus is on the effects of regulatory mechanisms in times of crisis. The paper analyzes a vast amount of fine-grained data on electricity prices for different countries over several years. For this, the paper employs an entropy-based measure.

Given the recent crises we are going through and the longer-term perspectives in the context of climate change, the topic appears highly relevant.

However, I have two interrelated concerns regarding the paper:

First, the paper's aim remains unclear: The paper comprises one sentence addressing the aim: on page 3, one reads “The paper aims to analyze the degree of prediction of energy, oil, coal, and gas prices.” From my understanding, this refers to the methods (in a broad sense) employed for forecasting. A question then would be in how far the proposed entropy-based measure allows for better predictions than alternatives. Apparently, the paper does not compare the performance of different metrics used for predictive purposes against each other.

This leads to the second concern: Rather it appears that the paper has a kind of hypothesis-testing research aim – namely, the hypothesis mentioned in the last paragraph of page 3 – according to which, roughly speaking, external interferences like price regulations result in a decrease of entropy of prices. However, it remains widely unclear why or how this hypothesis comes about. Moreover, one may even argue that it is somewhat “self-evident” that mechanisms for regulating prices would reasonably reduce the entropy of prices – or in other words: that regulators know effective mechanisms for regulation. Is it this that the paper wants to study?

Hence, I recommend clarifying the research objective and then redesigning the paper accordingly.

Reviewer #2: The paper addresses the challenges of forecasting energy prices, particularly in the context of recent global events such as the COVID-19 pandemic and geopolitical conflicts like the war in Ukraine. Traditional methods for analyzing energy prices, including statistical and econometric approaches, often assume stationarity and rely on models that can become inadequate due to the highly volatile and unpredictable nature of energy markets. The authors propose the use of entropy analysis, specifically Shannon entropy with the James-Stein shrinkage estimator, as a more effective tool for understanding and predicting energy prices.

Although interesting, the work presents some shortcomings. Primarily, the authors do not formally clarify which traditional methods their proposal serves as an alternative to, nor do they explain the disadvantages of using traditional methods in terms of the characteristics of the estimates (bias, efficiency, reliability, etc.). Secondly, the methodology used for estimation is taken for granted, leaving the reader in the position of having to consult the literature even for basic definitions. Finally, the work presents estimates and estimators, but it does not provide any indication of which measures were used regarding the reliability of the estimator, and thus the reliability of the obtained estimates.

In my opinion, the work aligns with the interests of the journal, but as indicated above, it requires some major revisions.

The use of English also needs revision, for instance, some expressions are too colloquial, and the verb tenses should be more consistent.

Below I list some suggestions:

Abstract, page 1, row 1: Remove "We" as the first word and use "the contribution" or "the paper" as the subject of the sentence.

Abstract, page 1, row 7: With reference to the last line (The work…entropy) and within the word limit for the abstract, specify in more detail the analytical tools used

Introduction, page 1, row 13: Include a definition of "probabilistic prices."

Introduction, page 2, rows 12-15: Rephrase the sentence “In normal times…monthly observation”; it is unclear which previous types of analysis are being referenced, nor the connection to the absence of random walk. Additionally, provide a formal definition of “normal time.”

Introduction, page 2, rows 16-17: Replace the sentence with: “In this contribution, the entropy analysis as a statistical method is used to analyze energy prices in markets.”

Introduction, page 2, row 23: Remove [12] and insert it at the beginning of the sentence

: As in [12] “Thermodynamics…ones”.

Introduction, page 2, row 29: Provide a definition of "system order."

Introduction, page 2, row 35: Explain what is meant by "degree of prediction." Since the authors state it as one of the objectives of the paper, it is important to clarify this point. Perhaps the authors are referring to some measure of prediction error? If so, include these measures in the "Results" section.

Introduction, page 4, rows 3-4: Replace “To verify our hypothesis, we used Shannon entropy with an estimator by James-Stein [17], which is the method that allows” with "To verify the hypothesis, the Shannon entropy with an estimator by James-Stein [17] is used. The method allows us..."

Introduction, page 4, row 7: With reference to the sentence “This method seems to be best adapted to our case,” specify which characteristics of the method make it most suitable for the case under study and which other methods it is preferable to. If this is stated, it is necessary to include at least one comparison between methods, conducted on the same data, that highlights gains at least in terms of estimation accuracy. Such a comparison should be announced in the Introduction and presented in the Results section.

Introduction, page 4, rows 9-10: Replace the sentence “We also conducted a robustness analysis of the results we received.” with “A robustness analysis of the results is also conducted,” but above all, indicate which tools were used to conduct this analysis. Include a table in the Results section with the results of the robustness measures.

Methodology, page 4, row 31: Replace “Our assumption is that n is fixed…” with “In our assumption, n is fixed…”.

Methodology, page 4, row 32: Explain what is meant by "natural units."

Methodology, page 4, row 34: Define what the "underlying probability mass function" is; perhaps include the definition in the sentence “According to some interpretation…” on line 25.

Methodology, page 5, row 2: Explain exactly what the authors' considerations are in support of the choice of the JS shrinkage estimator.

Methodology, page 5, row 3: Explain what is meant by “averaging two different models.” Do the authors mean that one model is used in which the equation includes elements already present in two other different models? Perhaps it is just the word “averaging” that is not appropriate.

Methodology, page 5, rows 10-12: Why is Shannon more suitable than Tsallis and Rényi? See the comment related to line 2 of the same page.

Methodology, page 5, row 15: Replace “stay” with “is.”

Methodology, page 5, row 16: Replace “especially” with “particularly.”

Methodology, page 5, row 18: Replace “operations” with “transformations.”

Methodology, page 5, row 20: What are the tools that the authors define as “conventional”? It is necessary to specify what is being referred to.

Methodology, page 5, row 22: In which cases do other methods fail to achieve stationary processes?

Methodology, page 5, row 28: Replace “Have a look at…” with “Considering…”.

Methodology, page 5, row 30: Replace “…characteristics. In a series like this, time-varying…” with “Considering…characteristics, time-varying…”.

Results, page 6, row 1: Replace “The research in the article is based…” with “The application proposed in this paper is based…”.

Results, page 6, rows 3-4: Replace “We used the daily data that cover the period from January …” with “Daily data from January…are used.”

Results, page 6, rows 6-11: Replace “For each country we constructed three two-dimensional sets of the country's electricity price versus oil, coal and gas price respectively. For each sets and each country we have estimated time-dependent Shannon entropy [18] using, as stated earlier the James Stein shrinkage estimator [17], with two year window, step was on day. To get cumulated results for Europe we have calculated average of estimated entropy for each of the day.” with “For each country, three two-dimensional sets of the country's electricity price versus oil, coal, and gas prices are constructed. For each set and each country, time-dependent Shannon entropy [18] is estimated using, as stated earlier, the James-Stein shrinkage estimator [17], with a two-year window, and the step is one day. To get cumulated results for Europe, the average of the estimated entropy for each day is calculated.”

Results, page 6, row 9: The JS shrinkage estimator is an estimator, so it is necessary to introduce a reliability index for the estimates or a measure of the estimator's precision (for example, the simple and well-known Mean Square Error, MSE) and present the results clearly and prominently.

Results, page 6, rows 12-16: Rewrite for clarity, possibly breaking it into multiple sentences.

Results, page 6, row 20: Replace “The moment…” with “The time” or “The period.”

Results, page 8, row 4: Insert “stating” before “…that during the crisis…”.

Results, page 8, rows 6-7: Explain the meaning of “…there could be significant problems with short-term forecasts.” What problems? In what particular situations?

Results, page 8, rows 17-18: The meaning of the sentence “The subsequent decline…regulations and restrictions.” is unclear. Do the authors mean that additional variables (of a categorical nature?) should be available to explain the decline, representing the introduction (and effect) of regulations and restrictions? If such variables were available, how could they be considered? Perhaps by modifying the equation on line 6 of page 5, or in the definition of the sliding windows?

Conclusions, page 8, rows 25-27: The sentence “It may be suspected…system of prices” expresses a concept already stated in the previous sections. Either elaborate on the sentence or remove it.

Conclusions, page 9, rows 4-5: Remove the sentence “It confirms our hypothesis.”

Conclusions, page 9, row 11: Remove "As you can see."

Conclusions, page 9, rows 13-14: At the end of the sentence, introduce examples of cases where econometric models (specifying the class of models: ARIMA, ARCH, GARCH, VAR, VECM, GLM, DLM, etc.) are impossible to estimate in the financial context referred to by the authors.

Additionally, include possible future developments of the proposed work.

6. PLOS authors have the option to publish the peer review history of their article (what does this mean?). If published, this will include your full peer review and any attached files.

Reviewer #1: No

Reviewer #2: No

---

## [Author Response · Author response to Decision Letter 0]

11 Nov 2024

Dear Sirs,

Thank you for allowing us to submit a revised draft of our manuscript. We appreciate the time and effort that you and the reviewers have dedicated to providing valuable feedback on our manuscript. We have incorporated changes to reflect most of the suggestions provided by the reviewers.

Reviewer 1

Thank you for your pertinent comments; the purpose and hypotheses were unclear. Following your suggestions, we have improved and clarified our purpose and hypotheses so that they are clear, and we have specified our scope of research.

Reviewer 2

Thank you for your insightful comments and the amount of work you put into making the language corrections, which we have considered. In addition, in accordance with the comments:

- we have corrected the research objective and hypothesis,

- we have explained the choice of method (Shanon entropy and why we do not use, for example, the Tsallis and Renyi methods – which are used in chemical and physical sciences, and Shanon derives from information theory) and the use of the Jems-Stein estimator and why they seem to be appropriate in this case

- we have added and expanded the definitions of terms and the methodology and/or corrected the colloquial or illegible formulations.

- we have expanded the robustness check

- we have corrected the conclusions and added future studies.

- Comparison of results using other methods, such as the Hurst coefficient, econometric methods (ARIMA, GARCH, etc.) and different methods of examining entropy is in the course of our research. We hope it will be presented in subsequent publications.

Sincerely,

Daniel Papla, PhD

---

## [Decision Letter · Decision Letter 1]

25 Nov 2024

Analysis of Entropy on the European Markets of Energy and Energy Commodities Prices

PONE-D-24-36364R1

Dear Dr. Papla,

We’re pleased to inform you that your manuscript has been judged scientifically suitable for publication and will be formally accepted for publication once it meets all outstanding technical requirements.

Kind regards,

Alessandro Mazzoccoli, Ph.D.

Academic Editor

PLOS ONE

Additional Editor Comments (optional):

Reviewers' comments:

Reviewer's Responses to Questions

**Comments to the Author**

1. If the authors have adequately addressed your comments raised in a previous round of review and you feel that this manuscript is now acceptable for publication, you may indicate that here to bypass the “Comments to the Author” section, enter your conflict of interest statement in the “Confidential to Editor” section, and submit your "Accept" recommendation.

Reviewer #2: All comments have been addressed

2. Is the manuscript technically sound, and do the data support the conclusions?

Reviewer #2: Yes

3. Has the statistical analysis been performed appropriately and rigorously? 

Reviewer #2: Yes

4. Have the authors made all data underlying the findings in their manuscript fully available?

Reviewer #2: Yes

5. Is the manuscript presented in an intelligible fashion and written in standard English?

Reviewer #2: Yes

6. Review Comments to the Author

Reviewer #2: The authors have addressed my previous suggestions, and in my opinion, the work is suitable for publication.

7. PLOS authors have the option to publish the peer review history of their article (what does this mean?). If published, this will include your full peer review and any attached files.

Reviewer #2: No

---

## [Editor Report · Acceptance letter]

2 Dec 2024

PONE-D-24-36364R1 

PLOS ONE

Dear Dr. Papla, 

I'm pleased to inform you that your manuscript has been deemed suitable for publication in PLOS ONE. Congratulations! Your manuscript is now being handed over to our production team.

Kind regards, 

on behalf of

Dr. Alessandro Mazzoccoli 

Academic Editor

PLOS ONE